# Personalised therapeutic management of epileptic patients guided by pathway-driven breath metabolomics

Kapil Dev Singh [1,2], Martin Osswald[3], Victoria C. Ziesenitz [1], Mo Awchi[1,2], Jakob Usemann [1], Lukas L. Imbach[3], Malcolm Kohler[3], Diego García-Gómez[4], Johannes van den Anker[1], Urs Frey[1,2], Alexandre N. Datta[1,5 ✉] & Pablo Sinues [1,2,5 ✉]

## Abstract

**Background** Therapeutic management of epilepsy remains a challenge, since optimal systemic antiseizure medication (ASM) concentrations do not always correlate with improved clinical outcome and minimal side effects. We tested the feasibility of noninvasive real-time breath metabolomics as an extension of traditional therapeutic drug monitoring for patient stratification by simultaneously monitoring drug-related and drug-modulated metabolites.

**Methods** This proof-of-principle observational study involved 93 breath measurements of 54 paediatric patients monitored over a period of 2.5 years, along with an adult's cohort of 37 patients measured in two different hospitals. Exhaled breath metabolome of epileptic patients was measured in real time using secondary electrospray ionisation–high-resolution mass spectrometry (SESI–HRMS).

**Results** We show that systemic ASM concentrations could be predicted by the breath test. Total and free valproic acid (VPA, an ASM) is predicted with concordance correlation coefficient (CCC) of 0.63 and 0.66, respectively. We also find (i) high between- and within-subject heterogeneity in VPA metabolism; (ii) several amino acid metabolic pathways are significantly enriched ($p < 0.01$) in patients suffering from side effects; (iii) tyrosine metabolism is significantly enriched ($p < 0.001$), with downregulated pathway compounds in non-responders.

**Conclusions** These results show that real-time breath analysis of epileptic patients provides reliable estimations of systemic drug concentrations along with risk estimates for drug response and side effects.

## Plain language summary

The clinical management of conditions such as epilepsy can be challenging. Each person with epilepsy responds differently to antiseizure medication and side effects are common. One approach to address this challenge is therapeutic drug monitoring (TDM), whereby levels of drugs are measured in the blood to follow the response to treatment. However, drug concentrations in the blood do not always reliably predict wanted and unwanted effects of a treatment. Here we show that a simple breath test can provide reliable estimates of circulating concentrations of a widely used antiseizure medication. In addition, the information contained in the breath signature enables us to identify which patients are likely to benefit from the treatment and which ones are likely to suffer from unwanted side effects. Our findings might help clinicians to decide how to treat people with epilepsy and to choose appropriate drug doses.

[1] University Children's Hospital Basel, University of Basel, Basel, Switzerland. [2] Department of Biomedical Engineering, University of Basel, Basel, Switzerland. [3] University Hospital Zurich, University of Zurich, Zurich, Switzerland. [4] Department of Analytical Chemistry, University of Salamanca, Salamanca, Spain. [5] These authors contributed equally: Alexandre N. Datta, Pablo Sinues. ✉email: alexandre.datta@ukbb.ch; pablo.sinues@unibas.ch

The concept of personalised medicine revolves around the idea of providing the most effective treatment with the least side effects for a given patient. In this context, the purpose of therapeutic drug monitoring (TDM) is individualising the dose to achieve maximum efficacy and, at the same time, minimise toxicity, for certain drugs with a narrow therapeutic window. Standard-of-care TDM is based on the measurements of plasma/serum drug concentration. TDM has obvious clinical benefits for patients and healthcare systems. However, it also has limitations. First, it relies on blood sampling to determine drug concentrations, which can be cumbersome to perform in infants and children. Second, drug concentrations can often not correlate with improved clinical outcome and/or minimal side effects, due to highly variable, patient-specific drug metabolism[1].

Epilepsy is a complex neurological disorder affecting around 50 million people worldwide characterised by recurrent unprovoked seizures[2]. However, treatment with one or more antiseizure medications (ASMs) allows roughly 70% of patients to live seizure free, but in the long run, 40% of those patients relapse and about 25% develop pharmaco-resistance[3]. As a result, the overall therapeutic management of epilepsy (especially, in paediatric patients) remains a challenge. Such individualised responses to medication with narrow therapeutic ranges calls for a more comprehensive phenotyping approach, beyond just monitoring systemic drug concentrations. Breath analysis has made substantial progress over the last decade by emerging analytical technologies such as secondary electrospray ionisation-high-resolution mass spectrometry (SESI–HRMS). Breath-metabolome analysis by SESI–HRMS offers a number of advantages, including noninvasiveness, short analysis time, wide metabolic coverage and capabilities to perform actual compound identification of the detected molecules (as opposed to other techniques such as chemical sensors). The latter is key to provide biochemical interpretations, hence gaining insights into the pathophysiology and drug-disease interplay. Over the last decade, a number of efforts have lifted this technology to transition from an interesting analytical platform to a standardised technique with real potential in clinical settings[4–7]. Based on prior work suggesting that this technology is capable of detecting drugs as well as drug-modulated metabolites in exhaled breath[8–11], we hypothesised that this would be the case in a clinical setting, whereby it might contribute to improved phenotyping of patients with chronic epilepsy requiring TDM. Here we show that such breath-metabolomics approach has potential to reliably predict blood levels of valproic acid (VPA, an ASM) and to offer an additional patient screening layer by providing scores for side effects and response to ASMs with minimal interference into routine clinical practice and patient invasiveness.

## Methods

**Participants**. In total, 66 paediatric epileptic patients (mean ± SD age, 10.7 ± 3.9 years; 37 males and 29 females, Supplementary Data 1) from the University Children's Hospital Basel (UKBB), under treatment with various ASMs requiring TDM per standard care were enroled in this study. Furthermore, we also used real-time breath data of 41 adult epileptic patients (mean ± SD age, 51.6 ± 17.1 years; 29 males and 12 females, Supplementary Data 1) from the University Hospital Zurich (USZ) to predict blood concentrations of total and free VPA. All subjects were under steady state of their ASMs at the time of measurements.

In the paediatric dataset (from UKBB) out of 123 attempted breath measurements (from 66 patients), 30 failed (Supplementary Data 1), whereas in the adult dataset (from USZ), out of 41 attempted measurements (from 41 patients; in USZ, there were no multiple visits from the same patient during study duration), four failed (Supplementary Data 1). The reasons being either (i) some

patients suffered from severe neurological impairment, preventing them to understand the instructions of the exhalation maneuver, or that the side effects would not allow them to perform the breath test (hence they are unable), (ii) some technical issue with instrument during patient visit, or (iii) in rare cases, clinical laboratory could not return blood concentration of (any) ASMs (Supplementary Fig. 1). This means our final dataset used in the study contained 93 paediatric measurements (from 54 subjects) and 37 adult measurements (from 37 subjects).

Paediatric measurements/patients were further annotated as follows: no such annotations were made for the adult dataset as it was only used to predict blood concentrations of total and free VPA.

Following the aetiologic classification of epilepsy by the international league against epilepsy (ILAE)[12], paediatric patients were divided into three groups. Group 1 consisted of patients with epilepsies of structural origin, group 2 consisted of patients suffering from genetic epilepsies, as well as epilepsies of unknown origin and finally, in order to differentiate them from genetic epileptic encephalopathies, and group 3 consisting of developmental and epileptic encephalopathies was created. None of the enroled paediatric patients were suffering from epilepsies of metabolic, immune, or infectious origin.

In order to assess the clinical outcome of ASMs, we further classified each data point for three categories: side effects, response to medication, and electroencephalography (EEG). Patients were labelled for each of these categories as classes I, II or III (see Supplementary Table 1 for class definitions). Side-effects questionnaire PESQ[13] (see Supplementary Table 2) was used to facilitate the side-effect comparison for epilepsies of different origins.

For the downstream prediction of clinical outcome, the dataset was subdivided as follows:

1. Side effects: "no side effects" (class I) vs. "side effects" (class II and III combined).
2. Response to medication: "responders" (class I) vs. "non-responders" (class II and III combined).
3. EEG: "normal" (class I) vs. "abnormal" (class II and III combined).

**Instrumentation**. The analytical platform employed for real-time breath analysis consisted of a SESI source (SUPER-SESI, Fossil Ion Technology, Spain) coupled to a HRMS (Q Exactive Plus, Thermo Fisher Scientific, Germany; Fig. 1a and Supplementary Fig. 2). The SESI ion source was fed with 0.1% ammonium formate in water solution, flowing (solution driving pressure 1.3 bar) through a 20 μm ID noncoated TaperTip silica capillary emitter (New Objective, USA) to generate electrospray. The settings of the SESI source were as follows: sheath gas flow rate 60, auxiliary gas flow rate 2, spray voltage 3.5 kV, capillary temperature 275 °C and S-lens RF level 55.0. Under these conditions, the nano-electrospray currents were typically in the range of 130–135 nA. SESI temperatures were set at 130 °C for the sampling line and 90 °C for the ion-source core.

All real-time breath mass spectrometry measurements were performed in full MS mode (scan range m/z 100–400, AGC target 1e6 and maximum injection time 500 ms) in both positive- (microscans 2 and resolution of 140,000 at m/z 200) and negative- (microscans 2 and resolution of 70,000 at m/z 200) ion mode. Q Exactive Tune software (version 2.9) was used to directly control MS for these measurements. The mass spectrometer was externally calibrated on a weekly basis using a commercially available calibration solution (Pierce™ Triple Quadrupole, extended mass range, Thermo Fisher Scientific, Germany) and

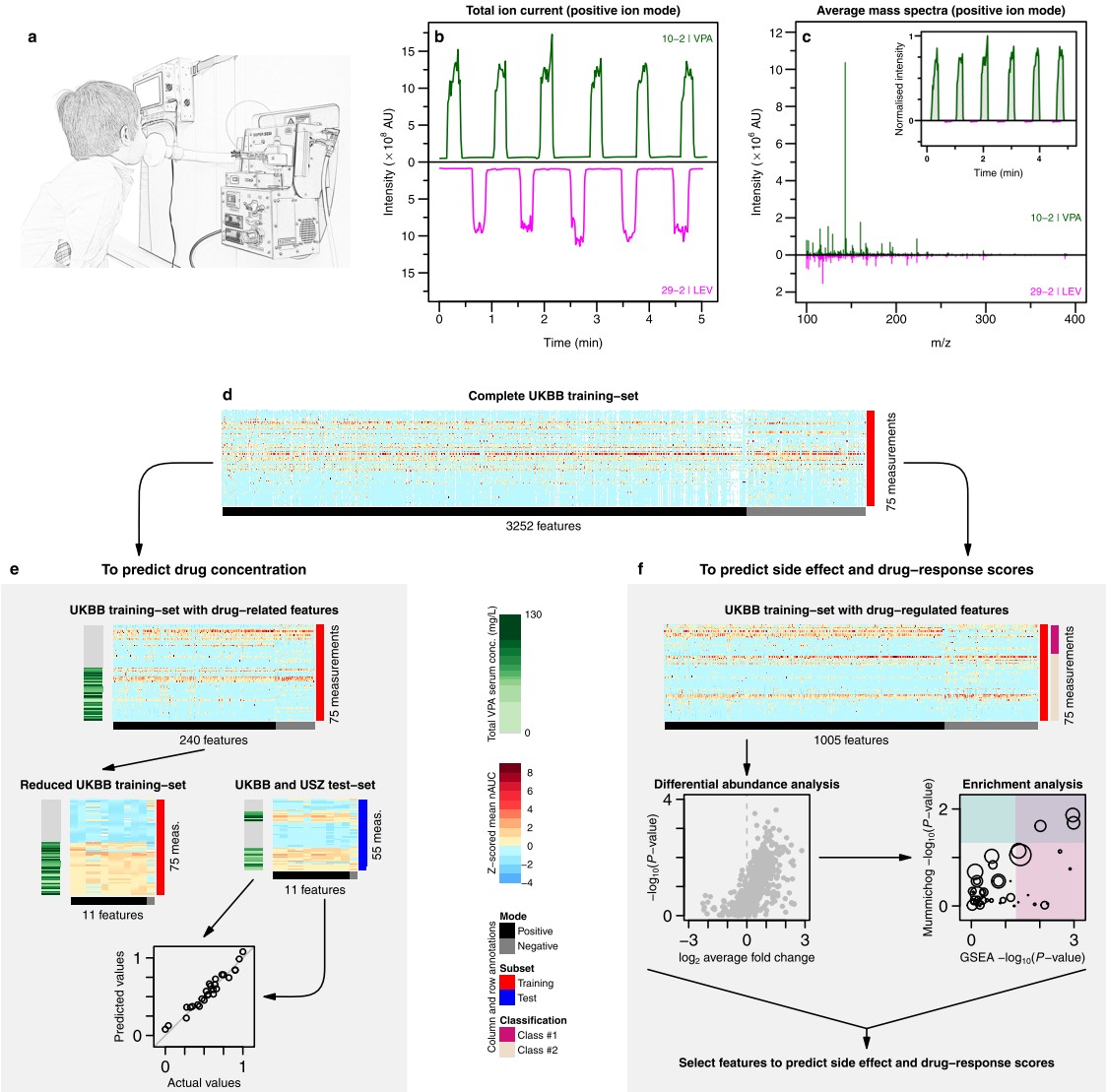

**Fig. 1 Overview of the study pipeline. a** The procedure begun with a patient performing five-to-six simple exhalations into a SESI–HRMS analytical platform located in the hospital premises. The breath metabolomics fingerprint was acquired in positive and negative-ion mode (5, 6 exhalations per mode). Shortly before the breath test, blood was drawn to evaluate blood/serum concentrations of ASMs. **b** SESI–HRMS is a real-time, noninvasive, and fast breath-metabolome analysis method. The whole breath test (i.e., positive- and negative-ion mode), lasts typically 10–15 min per patient. Positive-mode TIC from two patients, one receiving VPA and another one receiving LEV, is shown as an example (TIC of patient 29 is inverted to ease visual inspection). **c** Comparison of the average mass spectra between the two subjects taking VPA and LEV. The inset shows an example of time-trace at m/z 143.1066 (mass spectrum and time-trace of patient 29 inverted to ease visual inspection). For each ion, area under the curve during each exhalation was computed (shaded regions) and normalised by the exhalation time (nAUC). Then, the nAUCs of 5, 6 exhalations were finally averaged to represent mean nAUC of the ion. **d** This resulted in a 75 × 3252 (measurements × mass spectral features present in at least 10% of total measurements and correlated with exhalations) data matrix (z-score is only used here to ease visual representation; actual downstream analysis was done on raw numbers). **e** Analysis workflow used to predict VPA serum concentration based on drug-related metabolites. **f** The workflow used to predict side effects and drug-response scores based on drug-regulated metabolites. See Methods for more detail about panels **e** and **f**. Colour key for heatmaps is shown in-between panels **e** and **f**.

internally calibrated by using common background mass spectrometric contaminant masses as lock masses (positive mode: m/z 149.02332, 279.15909, 355.06993, 371.10123, and 391.28429; negative mode: m/z 60.99312, 73.0295, 87.04515, 89.02442, 101.0608, 115.07645, 225.23295 and 283.26425).

Serum concentrations of ASMs were measured at the clinical chemistry laboratory of University Hospital Basel (USB) as per their standard operating protocol (Supplementary Table 3).

**Procedures**. All subjects and/or parents, whichever applicable, signed informed consent to participate in the study in the

presence of their neurologist. This study was approved by the Ethics Committee of North–western and Central Switzerland (ID 2017-01537; see supplementary information for complete clinical protocol) and the Cantonal Ethics Committee Zurich (ID 2019-00030). The sample-size calculation included in the clinical protocol is shown in Supplementary Fig. 3. Subjects performed prolonged exhalations directly into SESI–HRMS system following blood draw for TDM (median = 21.2 min; IQR = 38.6 min). Figure 1a shows a representation of a child exhaling into the device (see Supplementary Fig. 2 for a bigger image). During each measurement, the subjects provided 5, 6 replicate exhalations, both in positive- and negative-ion mode (Fig. 1b and Fig. 1c). The

total time spent on the breath test was typically in the range of 10–15 min.

For VPA compound-identification purposes, we collected exhaled breath condensate (EBC) from one patient using an in-house condensation apparatus (containing dry ice and isopropanol). Collected EBC and pure standard of suspected molecules (dissolved in water) were analysed by ultra-high-performance liquid chromatography (UHPLC) system (Vanquish, Thermo Fisher Scientific, Germany) connected to HRMS. Samples were separated on a 50 °C heated pentafluorophenyl (PFP) column (Raptor FluoroPhenyl, 1.8 μm, 150 × 2.1 mm, Restek, USA) at a flow rate of 0.240 ml/min and eluted with a gradient between solvent A (water with 0.1% FA) and solvent B (methanol with 0.1% FA). The gradient profile was 50% solvent B between 0 and 1 min, 50–54% solvent B between 1 and 5 min, 54–95% solvent B between 5 and 5.2 min and 95% solvent B between 5.2 and 8 min followed by column reequilibration to 50% solvent B in a total 10 min run. Mass spectrometer was operated in positive-polarity full MS mode (scan range m/z 100–400, AGC target 1e6, maximum injection time 200 ms and resolution of 140,000 at m/z 200) triggering MS/MS acquisition (AGC target 1e6, maximum injection time 100 ms, resolution of 70,000 at m/z 200, loop count 5, isolation window 0.4 m/z and normalised collision energy 30) if it detects signal higher than 5000.

**Data analysis**. Raw mass spectra data from paediatric training-set patients were converted into mzXML file format using Proteo-Wizard's msConvert (version 3.0.11233) and imported into MATLAB (version 2019b, MathWorks Inc., USA) for further analysis. First, each spectrum from all files was aligned using the RAFFT algorithm implemented in MATLAB[14]. Then MATLAB's *mspeaks* and *ksdensity* functions were used to appropriately pick and extract the final list of 3252 features. These features were present in at least 10% of all measurements (to avoid noisy features) and were correlated with exhalations ($\rho_{spearman} >= 0.6$ and FDR <= 0.01) in each measurement (to avoid non-breath-related features). Finally, the mean of area under the curve during each exhalation normalised by exhalation time (nAUC) was computed for each of these features in all measurements (Fig. 1c). This resulted in a data matrix of 75 × 3252 (measurements × mass spectral features; Fig. 1d and Supplementary Data 2). This data matrix was then used to develop models (i) to predict drug concentrations (Fig. 1e) and (ii) to predict side effect and drug-response scores (Fig. 1f).

For VPA-concentration prediction (Fig. 1e), first, the full training-set was reduced to 240 features, which were present in at least 80% of the measurements, whereby the patients were receiving VPA (i.e., drug-related features). Later features in the training-set were further reduced to only 11 VPA-related features (Supplementary Table 4). Afterward, time-traces for these 11 features were directly extracted from all paediatric and adult measurements using in-house C# console app based on RawFileReader (version 5.0.0.38), an open-source.Net assembly from Thermo Fisher Scientific. These time-traces were then used to generate nAUC and three different matrices (UKBB training-set, UKBB test-set and USZ test-set). The *ComBat*[15] function from sva (version 3.34.0)[16] was then used to remove the known batch effect from these matrices (Supplementary Fig. 4 and Supplementary Data 2). This reduced training-set was finally used to screen for the best regression model (see Supplementary Figs. 5 and 6). Finally, we found Gaussian process regression using exponential kernel (i.e., eGPR) to be best performing on the training-set and hence it was used on an independent test-set containing paediatric and adult patients for final predictions.

In order to gain further insights into the rest of the metabolic signature captured in breath (Fig. 1f), first, the full training-set was reduced to 1005 features present in at least 50% of total measurements and with a CV greater than 30% (i.e., drug-regulated features). Later, two-sample *t*-test was performed followed by false-discovery rate (Supplementary Fig. 7). Afterward, MetaboAnalystR (version 2.0.4)[17] was used to add more biological insights into differentially abundant ions, by translating ions to metabolic pathways. The prediction of side effects vs no side effects and non-responders vs. responders in the training-set was conducted using significant metabolites identified by the enrichment analysis using first-principle-component (PC1) score. On this score, using only training-set data, a cutoff was assigned (based on Youden's index) to separate predicted classes (Supplementary Fig. 8). Later, we projected UKBB test-set data on the training-set PC1 score to complete this analysis.

**Reporting summary**. Further information on research design is available in the Nature Research Reporting Summary linked to this article.

## Results

**Overview of study pipeline and participants**. During the course of this study, whenever blood-based TDM was performed as per standard of care, the epileptic patients were asked to provide a breath sample (Supplementary Fig. 2). Figure 1 shows the overview of the study pipeline (see Methods section for more details). In total, 93 successful measurements from 54 paediatric subjects (Supplementary Data 1) covering a wide range of clinical presentations and pharmacotherapies, were performed at the UKBB (Fig. 2 and Supplementary Figs. 9 and 10). This group of patients is a representative real-life sample of unselected hospital out-patients. In addition to paediatric test-set measurements, we also used real-time breath data of 37 adult epileptic patients (Supplementary Data 1) from the USZUSZ to independently predict blood concentrations of VPA, a well-known ASM.

**Predicting drug concentration using drug-related metabolites**. VPA was the most prescribed drug in our cohort (50 out of 93 measurements involved VPA either as monotherapy or in combination with other drugs, see Supplementary Fig. 10), which prompted us to subdivide our dataset into a training and test-set to develop a regression model to predict VPA serum concentration based on a reduced number of breath signals (Fig. 1e). The reduced number of predictors consisted of 11 mass spectral features (Fig. 3a, b and Supplementary Table 4), which, upon further laboratory investigation (i.e., UHPLC–MS/MS of EBC; Fig. 3c–f and Supplementary Data 3) and also according to the literature[18–20], were found to be, as expected, stemming from VPA. The 11 features were assigned to four unique molecules: VPA itself and three metabolites (see VPA metabolic pathway in Supplementary Fig. 11a based on[18,19,21]). Namely, (i) 3-heptanone, which is a nonenzymatic end product of the $\beta$-oxidation pathway of VPA; (ii) 4-OH-$\gamma$-lactone, which is an end product of the $\omega_1$-oxidation VPA; (iii) a third metabolite with molecular formula $C_7H_{12}O_2$. Importantly, 4-OH-$\gamma$-lactone has been unambiguously identified now, as previous identifications were based on MS/MS only that could not resolve between this lactone and a different potential isomer (i.e., 4-ene VPA), which shows very similar fragmentation pattern (Fig. 3f). In addition, to the best of our knowledge, $C_7H_{12}O_2$ is a novel VPA metabolite not reported in the literature. Based on chemical reasoning and comparison with known VPA-degradation pathways, we hypothesise that it could be either 2,3- or 2,5-heptanedione (Supplementary Fig. 11a; hereafter referred as heptanedione). As expected, even by simple visual inspection of the breath mass

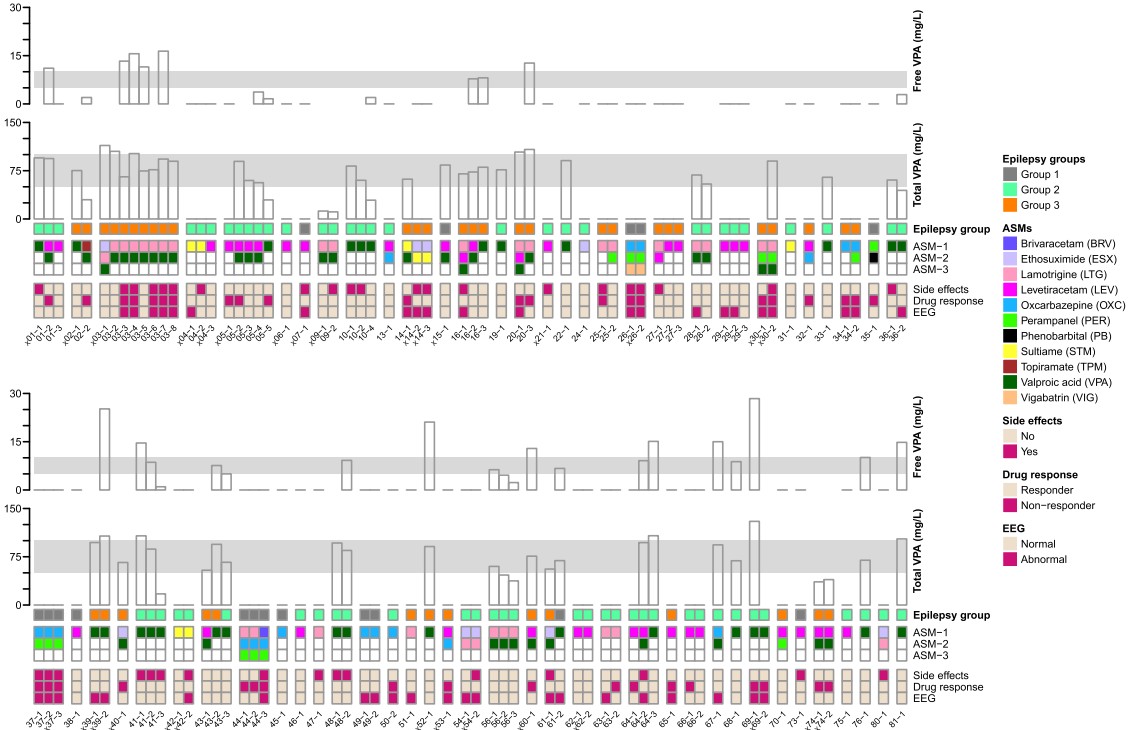

**Fig. 2 Overview of studied paediatric patients.** We screened a heterogeneous group of patients coded in the figure as patient ID-visit number (× denotes measurement with unsuccessful breath test). The patients suffered from epilepsies of different origins, as indicated in the row epilepsy group. Group 1 consisted of patients with epilepsies of structural origin, group 2 consisted of patients suffering from genetic epilepsies, as well as epilepsies of unknown origin, and group 3 consisted of developmental and epileptic encephalopathies (DEE). The range of pharmacotherapies is described in ASM rows. The patients were receiving either a single drug or combinations of up to three drugs. The bottom three rows identify the clinical outcome as described by side effects (no or yes), drug response (responder or nonresponder) and EEG (normal or abnormal). The hollow vertical bars on the top show the concentrations (mg/L) of free and total serum VPA (level 0 was assigned to all the measurements when the subject was not taking VPA, missing bars in free VPA represent the unavailability of data). Grey horizontal bands indicate the therapeutic range of VPA (i.e., 50–100 mg/L for total and 5–10 mg/L for free VPA). Thus, for example, patient 20 suffers from DEE. During the first visit (i.e., 20-1), the patient was receiving a combination of LTG, LEV, and VPA. Total serum VPA concentration was 104 mg/L (i.e., beyond the therapeutic range) and free serum VPA could not be measured. At this point in time, the patient had no side effects, but was not responding to the medication, and EEG was abnormal. On the second visit, the patient did not perform breath test. During the third visit (i.e., 20-3), LEV had been removed, total serum VPA concentration was 108 mg/L, and free serum VPA concentration was 12.7 mg/L (both levels above the therapeutic range). There were still no side effects, EEG was back to normal, but the patient was still not responding to the pharmacotherapy.

spectra, one can appreciate that the signal intensity of these 11 mass spectral peaks was overwhelmingly more abundant in the patients taking VPA than in the patients taking other ASMs (Supplementary Fig. 12). However, the differences were less obvious for heptanedione and VPA molecules because non-VPA patients exhale other endogenous compounds (e.g., octanoic acid) that are isomers (i.e., the same exact mass) and hence cannot be resolved by SESI–HRMS[22].

We further trained a regression model based on Gaussian process regression using exponential kernel (i.e., eGPR, see Supplementary Figs. 5 and 6 for details about how this model was selected) to predict the total and free serum VPA concentration, based on the signal of these 11 ions detected in exhaled breath. Figure 4 shows the predicted total and free VPA serum concentrations against the actual serum concentrations for the paediatric and adult population test-set (also see Supplementary Data 4). We used Lin's concordance correlation coefficient (CCC) to evaluate the agreement between actual and predicted serum concentrations[23]. To build a complete and accurate model, we included patients receiving other drugs but VPA to capture the whole range of concentrations from zero because some VPA-taking patients may actually be well below the therapeutic range. Supplementary Fig. 13 clearly shows that the model predicts accurately the zeros (i.e., patients not receiving VPA). Regarding

the prediction of VPA patients (i.e., real-world scenario, Fig. 4), the model could predict reasonably well the systemic total VPA concentrations (CCC of 0.63). For example, in the therapeutic range of total VPA (i.e., 50–100 mg/L), the prediction was reasonably accurate; however, some patients were clearly under- or overpredicted (especially those outside the therapeutic range). One explanation for such deviation is that exhaled VPA (and its metabolites) should mirror the free fraction of VPA (rather than the total VPA), as protein-bound VPA cannot be detected in breath and only the free fraction will undergo further metabolism. Despite the limited data availability (mostly because of the difficulty to acquire enough blood sample for free VPA quantification), Fig. 4b shows better free VPA prediction for the paediatric dataset (CCC = 0.84).

**Predicting side-effect and drug-response scores using drug-modulated metabolites.** The second branch of our analysis (Fig. 1f) aimed at identifying endogenous metabolites altered in the training-set measurements of patients suffering from side effects (23/75), or not responding to pharmacotherapy, i.e., non-responders (26/75), or those showing abnormal EEGs (27/75) on the day of consultation (Supplementary Fig. 14). We observed a general trend of exhaled breath metabolites to be (i) upregulated

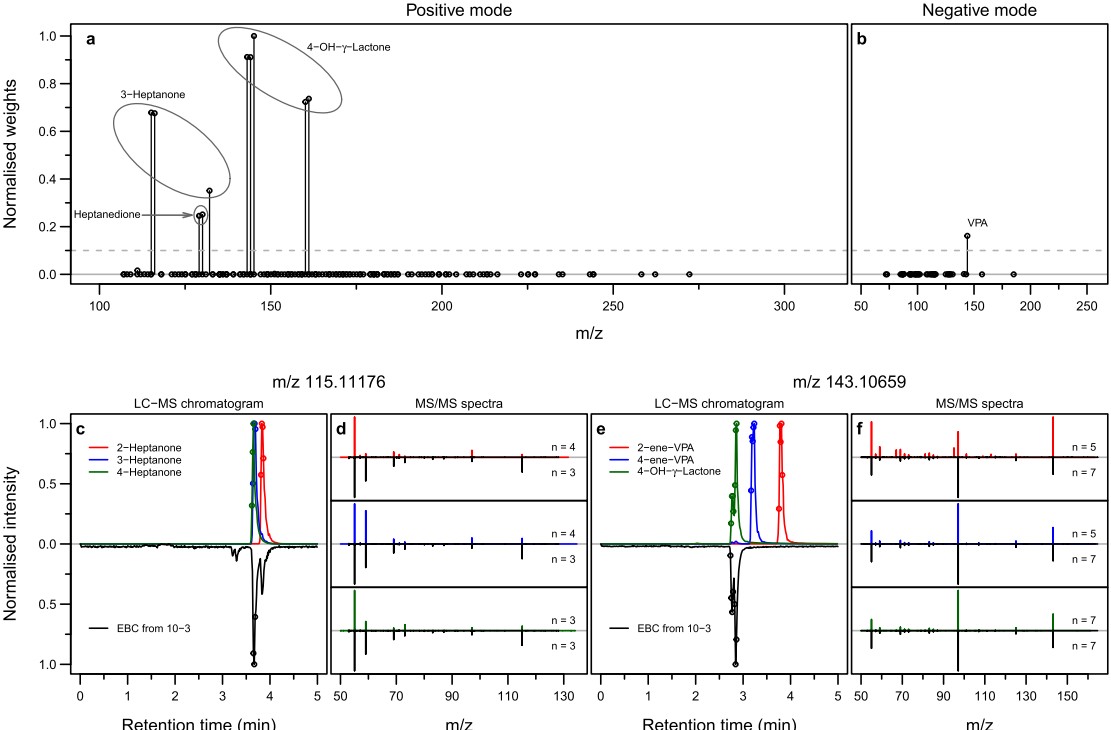

**Fig. 3 Selected predictors of importance belong to four distinct molecules. a, b** Based on training-set and the total VPA serum concentration, the contribution of each ion (predictor) was assessed by using MATLAB's *relieff* and *TreeBagger* functions. Predictor importance from both methods was combined to generate an overall normalised weight assigned to each predictor from positive (**a**) and negative (**b**) modes. Predictors with weight above an empirical cutoff of 0.1 (shown by dashed grey line) were selected for the next steps. **c–f** Show the compound identification via LC–MS, based on the combination of retention time and MS/MS spectra match. We confirmed that nominal m/z at 115 belongs to 3-heptanone (**c**, **d**) and nominal m/z at 143 belongs to 4-OH-γ-lactone (**e**, **f**). This figure shows the comparison of LC–MS chromatograms (**c** and **e**) and the average MS/MS spectra (**d** and **f**) between the pure standards of suspected molecules and EBC from a VPA taker (subject 10, 3rd visit). Number of averaged MS/MS spectra is denoted by *n* and the time at which those MS/MS spectra were obtained is denoted by the open circles in the corresponding LC–MS chromatogram.

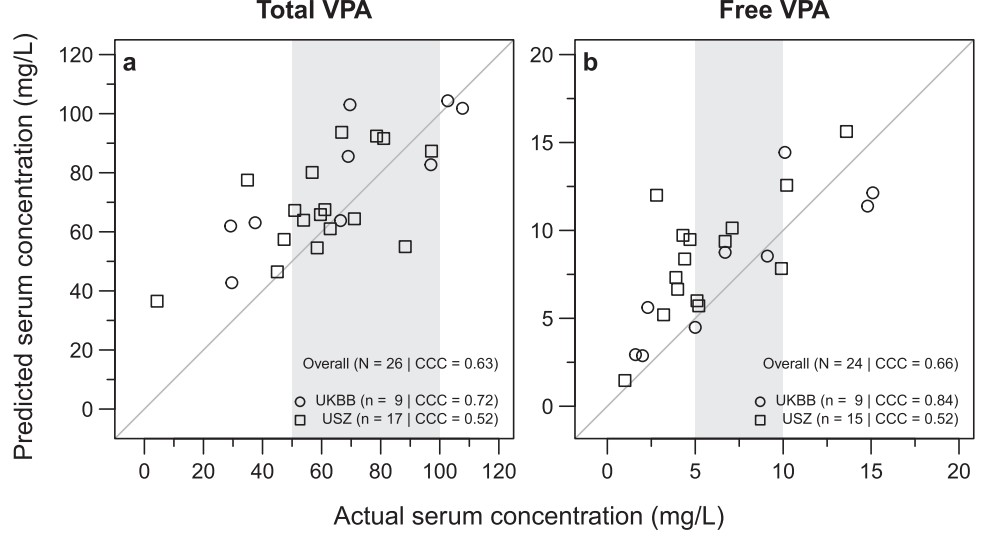

**Fig. 4 Prediction of systemic drug concentration based on real-time breath mass spectra.** Prediction of total (**a**) and free (**b**) VPA serum concentration of independent test-set containing paediatric subjects from UKBB and adult subjects from USZ. Vertical grey box shows the reference therapeutic range of 50–100 mg/L for total VPA and 5–10 mg/L for free VPA. Solid grey line represents the identity (y = x) line.

in children suffering from side effects as compared with no side effects and (ii) downregulated in non-responders than in responders (Supplementary Fig. 7). In contrast, abnormal EEG showed no significant change in the levels of exhaled metabolites as compared with normal EEG. Subsequent pathway-enrichment analysis using MetaboAnalystR[17], revealed significant enrichment ($p < 0.01$) of several amino acid metabolic pathways (Fig. 5 and Supplementary Data 5) in patients suffering from side effects. Whereas, only tyrosine metabolism was found to be significantly enriched ($p < 0.001$) in non-responders (Fig. 5 and

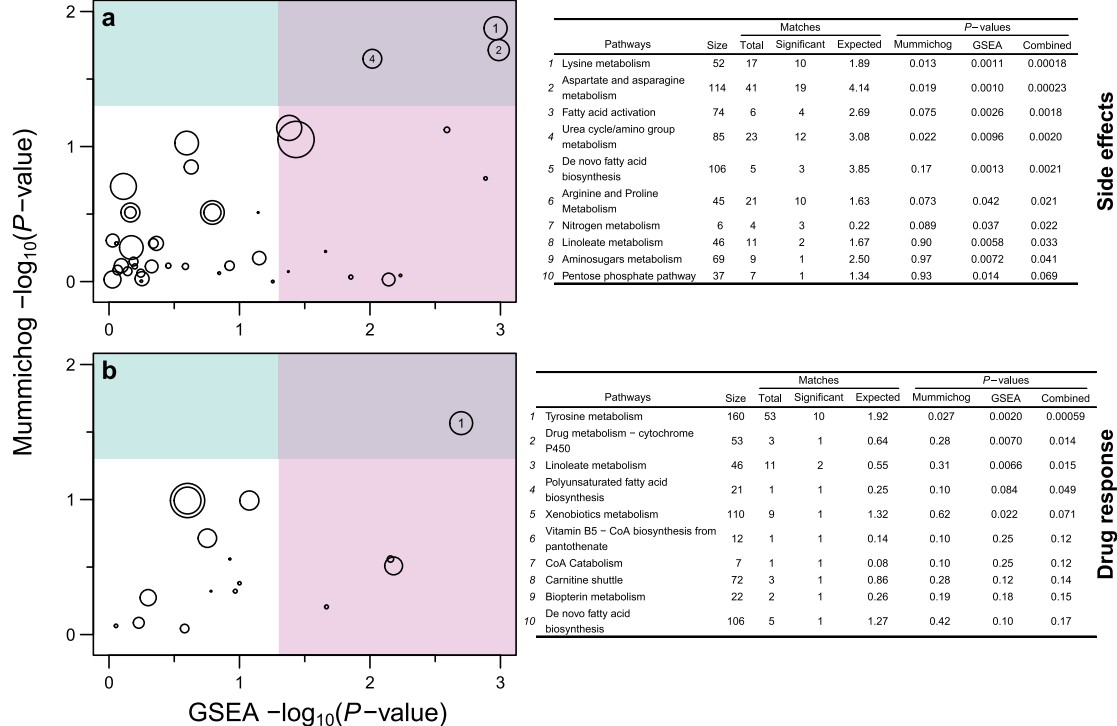

**Fig. 5 Association between differentially abundant ions and metabolic pathways. a, b** MetaboAnalystR was used to assign metabolic pathways to differently abundant ions for side effects (**a**) and drug response (**b**). The figure shows the scatter plot for altered pathways (shown by open circles) based on two different algorithms. Size of each pathway circle is proportional to the $\log_2$-scaled fold enrichment/depletion of the pathway. The area within the blue horizontal box represents Mummichog $P$-values less than 0.05, whereas the area within the red vertical box represents GSEA $P$-values less than 0.05. We considered pathways with $P$-values less than 0.05 in both algorithms (top-right quadrant) for further analysis. The number inside the circles in the top-right quadrant corresponds to pathway rows in the adjacent tables (only top 10 rows are shown).

Supplementary Data 5). Figure 6a zooms in the altered compounds identified by these analyses (see Supplementary Data 6 for further details about altered compounds). To further complete our analysis, we explored whether the endogenous altered metabolites could be used to predict which patients are likely to respond to pharmacotherapy and to suffer from side effects. As our UKBB test-set contains a limited number of side-effect cases and non-responders, we used the whole UKBB dataset for this prediction (Fig. 6b and c and Supplementary Data 6). Finally, based on the results presented here, we proposed a clinical decision-making workflow based on real-time breath analysis (Fig. 7).

## Discussion

In this translational study, by combining real-time, noninvasive and rapid breath analysis with sophisticated bioinformatics tools, we showed that systemic VPA concentrations can be accurately predicted (Fig. 4). Earlier work with exhaled breath measurements of 3-heptanone and 4-OH-γ-lactone showed promising results regarding the use of these molecules as potential breath-based markers for therapeutic monitoring of VPA[19,20,24]. However, none of those studies performed any independent prediction of blood concentration of VPA. We also confirmed that separately protonated 3-heptanone (CCC of 0.06 and −0.16 in total and free VPA, respectively) and 4-OH-γ-lactone (CCC of 0.40 and 0.19 in total and free VPA, respectively) underperform (Supplementary Fig. 15a–d) compared with our proposed model with 11 predictors (CCC of 0.63 and 0.66 in total and free VPA, respectively). Interestingly, we observed that including covariates such as age, gender and number of ASMs as predictors, does not necessarily make the VPA prediction any better (Supplementary Fig. 15e and f). Hence, to reduce the complexity, we used the

model with only exhaled VPA-related ions as predictors. Notably, the signal-intensity distribution for these drug-related ions was similar across adult and paediatric populations (Supplementary Fig. 16). For this reason, we attempted to predict also adults' VPA concentrations using the model created with children (Fig. 4). Not surprisingly, the model performed better predicting the children's subpopulation (free VPA CCC = 0.84 for children vs. 0.52 for adults). More accurate predictions are expected when creating a regression model using an adult population as training dataset. Overall, these results show the feasibility of estimating systemic VPA via breath analysis in a clinical context.

It has also been shown that free VPA is physiologically active and clinically relevant, which stresses the importance of measuring free VPA concentration[25,26]. In spite of this, current clinical practice relies most often on total VPA blood levels, perhaps due to the fact that determination of free VPA requires relatively large blood volumes, lengthy and laborious mass spectrometric analyses requiring hours-to-days of laboratory work (Supplementary Table 3). In our paediatric dataset, 9 out of 42 (21%) requests to determine free serum VPA failed, seven of them due to lack of enough material (Supplementary Data 1). Additionally, we observed that keeping total VPA within therapeutic range does not guarantee that ultimately free VPA will also do so (Supplementary Fig. 17).

Perhaps even more importantly, conventional TDM presents fundamental shortcomings. The lack of clinical correlation between the efficacy outcomes or side effects with ASM concentrations due to high inter- and intra-individual variation, decreases the value of TDM. This limitation discourages practitioners to use TDM, except perhaps in specific circumstances (e.g., pregnancy and known pharmacokinetic interactions). The limitation that keeping the drugs' concentrations in the

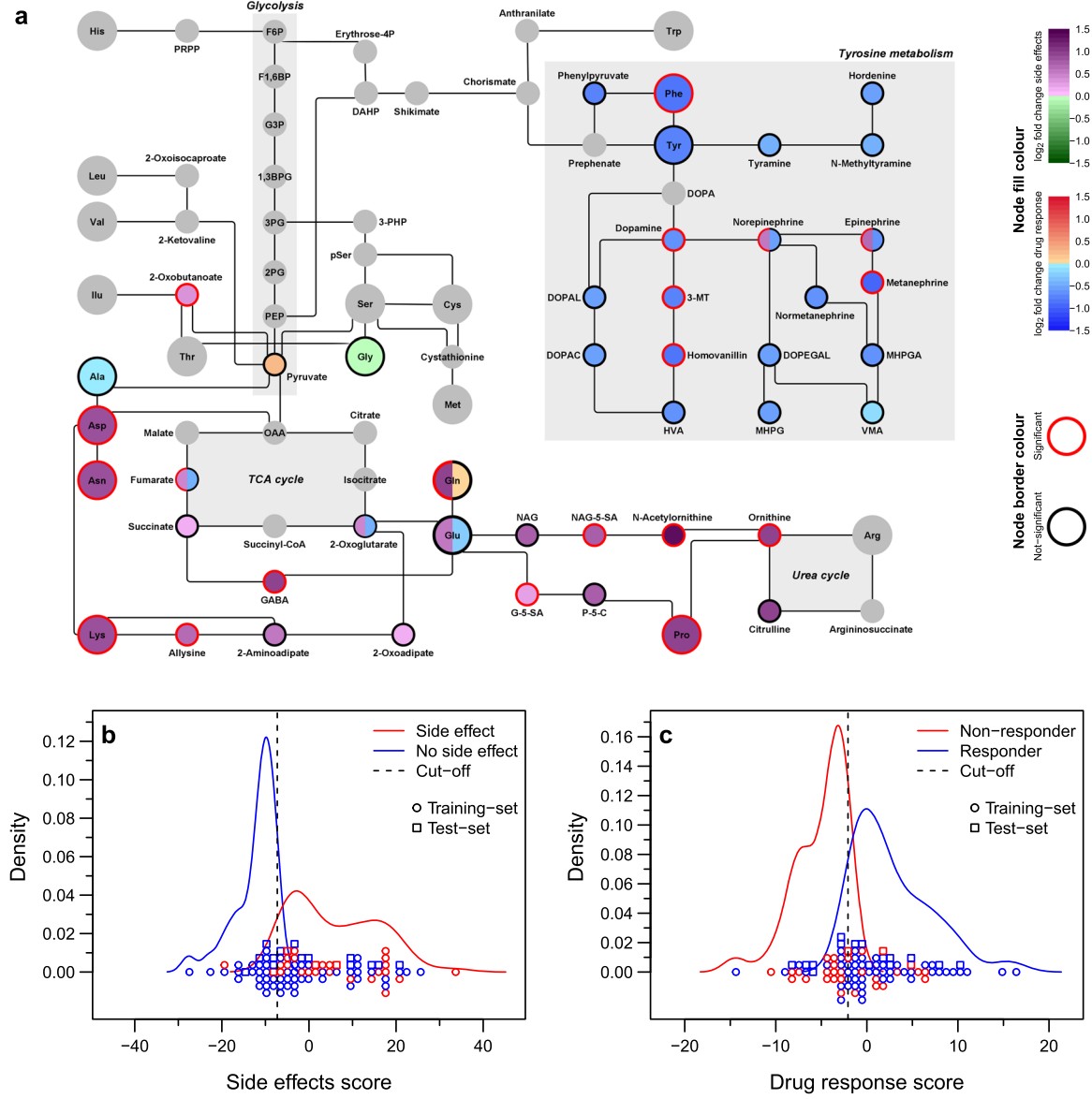

**Fig. 6 Real-time breath analysis offers a noninvasive window into altered metabolic pathways in patients not responding to pharmacotherapy and/or suffering from side effects.** a SESI–HRMS breath analysis detected alteration in the levels of several amino acids and associated compounds in epileptic patients. The figure shows a simple (unweighted, undirected, no loops, or multiple edges) graph of amino acid metabolism (based on KEGG map01230: biosynthesis of amino acids). Each node is a compound, where node-fill colour represents the mean $\log_2$-scaled fold change in the side effects (yes vs. no), and drug-response (nonresponder vs. responders) dataset from the training set. Node-border colour represents whether the compound was assigned to significant or background list via MetaboAnalystR. Only compounds from the significantly enriched pathways (top-right quadrant in Fig. 5) are coloured. The rest are shown as grey. Node with two colours (node split) denotes that the compound was present under significantly enriched pathway of both datasets. **b, c** Density plot for the predicted score and classes of having side effects (**b**) and drug response (**c**). Density curves are accompanied with the actual data points, where each point represents one measurement from UKBB dataset, coloured based on clinically observed side effects (**b**) and clinically observed drug response (**c**). On predicted scores, a cutoff was assigned (based on Youden's index calculated using only training-set data) to separate predicted classes.

therapeutic range, does not assure an optimal clinical response of no side effects, was also evidenced in our cohort. For example, in 32% of the paediatric measurements (40 out of 123), the patients were not responding to the therapy. However, of these 40 measurements, only in 18, at least one drug was outside its reference range.

Similarly, in 37 paediatric measurements, the patients were suffering from side effects. However, blood concentration of at least one drug was outside of its reference range in only 14 of these cases. Take for example visits 5 and 6 of subject 3 (Fig. 2 and Supplementary Data 1). In both visits, the patient was under

the same therapeutic regime, with similar total VPA and lamotrigine (LTG) levels in blood. However, while the child was responding properly, not suffering from side effects and presented normal EEG in visit 5, in visit 6, the situation was dramatically reversed. Such individual-specific response to the medication further stresses the importance of adopting a more comprehensive and personalised stratification approach.

Here we showed the possibility of predicting free VPA concentrations in 15 min by a simple noninvasive breath test. But perhaps more importantly, the prediction is based on VPA and its downstream metabolites; hence, further insights could be gained

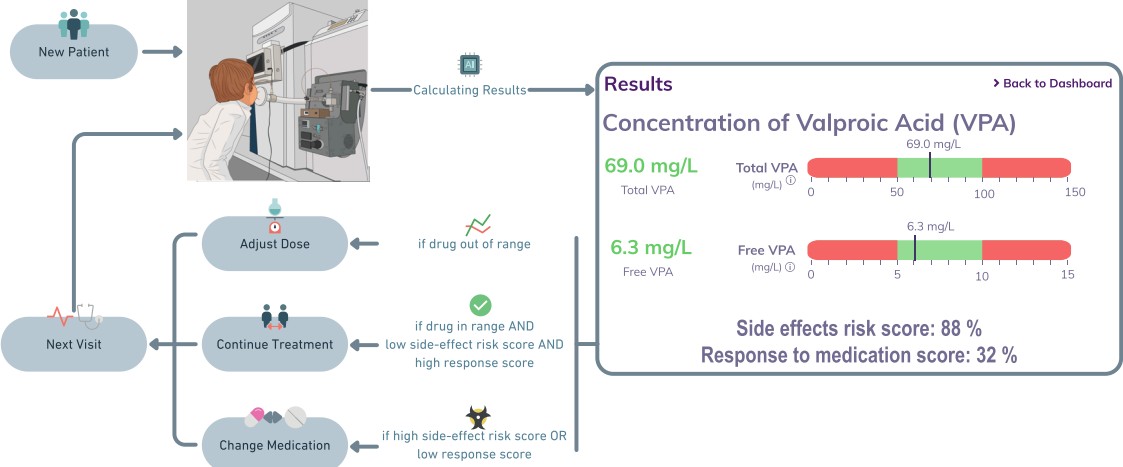

**Fig. 7 Proposed concept of extension of current TDM methods to monitor noninvasively drug-related and drug-modulated metabolites in exhaled breath.** The workflow starts with screening the breath metabolome of a new patient requiring TDM. Then, drug serum concentrations are predicted based on drug-related metabolites, whereas drug-modulated metabolites can be used to estimate side effect and drug-response scores. This information is available within 15 min on the same day of consultation. Based on this information, the responsible clinician decides whether to continue treatment, adjust the dose, or change medication. Once the decision is made and implemented, the patient's breath metabolome is reassessed in the next visit to repeat the whole process.

on how the drug is metabolised at an individual level. For example, based on 3-heptanone and 4-OH-$\gamma$-lactone signal-intensity ratios, we could estimate inter- and intraindividual variability in the activity of $\beta$- and $\omega_1$-oxidation pathways of VPA metabolism. We found considerable heterogeneity in metabolic pathway patterns within the sampled population and even within subjects during the course of this study (Supplementary Fig. 11b).

Moreover, because the breath test captures a large number of metabolites (well beyond the drug-related metabolites just described), we explored whether some of these may be associated with the actual clinical outcome. We observed a significant enrichment ($p < 0.001$) of the tyrosine metabolism, where pathway compounds were downregulated in the subset of patients not responding to medication (Fig. 6a and Supplementary Data 6). The association between downregulation of tyrosine metabolism and increased number of seizures (i.e., not responding adequately to medication) may be rationalised by the fact that neurotransmitter dopamine, which is known for its anti-epileptic action[27], is also downregulated, since it is synthesised from tyrosine and phenylalanine[28]. In addition, previous studies have suggested that administration of D-leucine[29], glutamic acid[30] and tyrosine[31], can reduce seizure frequency, which also correlates with detecting lower levels of glutamic acid and tyrosine in the non-responders of this study. Additionally, multiple earlier studies have reported on the alteration in the amino-acid concentrations in the cerebrospinal fluid and the plasma during epilepsy[32,33]. Previously we have also shown that, at least for some of the amino acids mentioned here, breath levels correlate with blood concentrations[34]. Moreover, recently several studies based on animal models and humans have reported about the effects of branched-chain amino acids on epilepsy[35] and the importance of having amino-acid-balanced diet[36]. Taken all these observations together, it seems worth exploring different intervention routes to rebalance lower amino-acid levels found in non-responders.

Furthermore, we also found a strong enrichment of urea cycle/amino acid metabolism, where pathway compounds were mainly upregulated in measurements with side effects (Fig. 6a and Supplementary Data 6). Previously, different randomised crossover trials have shown that glycine improves sleep quality[37], which correlates with our observation of detecting lower levels of glycine

in patients with side effects (such as somnolence). Additionally, several other amino acids, such as aspartic acid, glutamic acid and $\gamma$-aminobutyric acid (GABA) have been shown to be increased in blood of subjects with aggressive behaviour[38], this is also consistent with observing a higher level of these amino acids in patients with side effects (such as irritability). VPA is known to increase blood levels of proline[39] and GABA[40]. Since the majority of our population were taking VPA, higher levels of proline and GABA observed in side-effects case can also be attributed mostly to VPA. Overall, we believe that these findings open new routes for metabolic pathway-guided drug monitoring and management, by providing risk estimates for side effects as well as drug-therapy effects (Fig. 6b and c)[41]. Pathway analysis may not only be relevant for the dosage of the ASMs, but also may potentially be useful for the choice of the ASMs (e.g., to rebalance amino-acid metabolism).

However, this study has several limitations. First, the inclusion criteria allowed for a quite heterogeneous patient population, which may explain the limited false-discovery rates found in the comparison between responders and non-responders. Second, chemical identification of endogenous metabolites associated with non-responders and side effects could only be postulated based on the database matching of measured accurate masses (within 2 ppm). Chemical identification with the highest degree of confidence would require further UPLC–MS/MS analysis using chemical standards, as done in this work for VPA metabolites. For this reason, although the enrichment analysis algorithm used here analyses at a collective level the behaviour of groups of metabolites (assuming random errors at the individual peak level), the biochemical/metabolic interpretation should be taken cautiously, until unambiguous chemical identification is provided. Third, the reported altered pathways in non-responders and patients suffering from side effects could not be completely tested in an independent cohort. Fourth, the prediction of (total and free) VPA concentrations in the adult dataset was done using regression model trained on paediatric dataset, due to limited size of the adult dataset. Most of these limitations can be overcome by ongoing recruitment of more participants; however, the results shown here provide sufficient evidence on the feasibility of applying breath metabolomics in such clinical context.

In conclusion, based on the evidence presented here, we propose that SESI–HRMS breath testing may serve as a companion diagnostic approach to potentially minimise drug side effects and help to choose the seizure-specific treatment. Key advantages that make it ideal for the hospital in- and outpatient setting include noninvasiveness and real-time results. It is thus ideally suited for chronically ill patients and children in real time during an outpatient's consultation. This, along with the ability to predict serum concentration of drugs, allows us to propose a clinical decision-making workflow based on real-time breath analysis (Fig. 7). Furthermore, we believe the data presented here will serve as the foundation that may transform today's antiseizure therapy selection approach to a more objective pathway-based personalised approach. Future randomised controlled drug trials may profit from SESI–HRMS breath metabolomics analysis at study entry to better characterise responding/nonresponding patients. We finally envision that this concept has potential to be deployed in drug monitoring of other chronic diseases.

## Data availability

All the data generated and analysed that support the findings in this study are within the article and its supplementary information files and are available from the corresponding author upon reasonable request. Additionally, the RAW and mzXML files of the real-time breath measurements are available from the MetaboLights (https://www.ebi.ac.uk/metabolights) repository (accession number MTBLS2400).

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

## Acknowledgements

We thank the participants who volunteered to take part in this study and the study nurses' team (Amelia Imolesi, Maya Weber and Isabel Gonzalez Novoa) for coordinating and maintaining study documents. Mélina Richard is gratefully acknowledged for her research coordination and help with Fig. 7. Dr. Nadine Oser is gratefully acknowledged for coordinating the recruitment of the participants. Prof. Katharina Rentsch of University Hospital Basel is greatly acknowledged for analysing the serum samples to determine the concentrations of ASMs. This work is part of the Zurich Exhalomics project under the umbrella of University Medicine Zurich/Hochschulmedizin Zürich. PS

received funding from Fondation Botnar (Switzerland) and the Swiss National Science Foundation (320030_173168 and PCEGP3_181300).

## Author contributions

K.S., V.C.Z., J.v.d.A., A.N.D. and P.S. designed the study. K.S., M.O. and M.A. did the breath mass spectrometric measurements. K.S. and P.S. developed the pipeline to analyse the breath mass spectrometric data. K.S. collected EBC and performed LC–MS/MS measurements. K.S., M.K., D.G.G., J.v.d.A., U.F., A.N.D. and P.S. interpreted the data. K.S., U.F., A.N.D. and P.S. wrote the manuscript. V.C.Z., J.U., L.L.I., M.K., D.G.G. and J.v.d.A. contributed further to the writing of the manuscript. A.N.D. and L.L.I. assessed the clinical status of the paediatric and adult epilepsy patients, respectively.

## Competing interests

PS and MK are cofounders of Deep Breath Intelligence AG (Switzerland), which develops breath-based diagnostic tools. KS is consultant for Deep Breath Intelligence AG (Switzerland). All other authors declare no competing interests. The breath-analysis concept of predicting drug concentrations and risk scores, and the data-processing pipeline discussed here have been incorporated in the European patents 20186274.5, filed on July 16th 2020 and 21185400.5, filed on July 13th 2021, respectively.
