## [Peer Review File · Communications Medicine]

Reviewers' comments:

Reviewer #1 (Remarks to the Author):

The present study is very relevant for improving clinical care of both children and adults with epilepsy suggesting a new method of non-invasive real time breath metabolomics to assess i) drug (specifically valproic acid) related metabolites and furthermore ii) drug regulated metabolites. I would categorize the present manuscript as a proof-of-concept study opening a possibility for a clinical breakthrough in evaluating individualized outcomes to a given anti-seizure medication (ASM – nowadays the preferred term). Naturally, much further research is needed to translate this new concept of measuring individualized effects with the help of this novel technique of not just valproic acid but other important ASM to everyday clinic, but the promise is definitely there and well enough backed by relevant analysis in this manuscript.

However, I would like to suggest to improve the line of reasoning both in the introduction as well as in discussion in order to better convey the full potential of this novel approach. The deficiencies in present day conventional therapeutic drug monitoring (TDM) should be made more clear. The lack of clinical correlation between the efficacy outcomes or side-effects with ASM concentrations really decreases the value of TDM due to huge inter- and intraindividual variation, and many clinicians use TDM sparingly for this reason only in specific circumstances (pregnancy, known pharmacokinetic interactions etc.). In this case the golden standard is really not so golden, but of course the predictive performance of the novel approach should be compared with the present standard. I don't think the need to correlate ASM effects with metabolic pathways is less important in adults compared with children.

The discussion about results concerning drug related metabolites and drug regulated metabolites should be dealt in a more precise way. Drug related metabolites are more straightforward and the emphasis on free concentrations of valproic acid should be made even more pronounced. The possibility of 3-heptanone and 4-OH-gamma-lactone with other predictors of blood valproic acid concentration to create new reference values independent of traditional blood reference values should be also explored. If any data from the present study or previously published studies is available, it would be interesting to have a short paragraph discussing the potential of this novel technique for measuring other ASM with this same approach.

The data about correlations between side-effects and efficacy outcomes with specific metabolites is exciting and opens a really important scenario for future clinical praxis. The possibility of measuring specific metabolites with predicting power for side-effects and efficacy would be a significant improvement truly revolutionizing epilepsy treatment paradigm. However, there is also a possibility that other factors besides drug regulated pathways have an effect on the concentrations of these different metabolites. Correlation does not necessary mean causation, individual differences in for instance gut-bacteria regulated amino acids concentrations may cause some individuals to have either high or low systemic or CNS concentrations prior to starting valproic acid and therefore the effect may depend on an individual's prior state to a varying degree. Therefore, new studies where the amino acid or other relevant molecule concentrations are measured prior to starting a new ASM or other therapies such as neuromodulation (VNS or DBS). This approach to my understanding is not just related to ASM regulation but has also other applications and should be also discussed.

The clinical data in present manuscript is well presented, and the inclusion of both adult and pediatric population is one of the strengths of the study. The limitations are well discussed.

Reviewer #2 (Remarks to the Author):

This very well written manuscript describes a detailed and original investigation into the real-time

breath profile of children and adults with epilepsy, with the aims of discovering potential breath biomarkers of drug exposure and efficacy, and also disease control (on symptoms and EEG) and drug side effects. The findings of a potential breath signature of valproate (levels and efficacy) are very intriguing and certainly merit further study.

I am not an expert in analytical chemistry or multivariate data analysis, but within these limitations I feel the methods and study design are sound and appropriate. I applaud the investigators on making the data open to independent scrutiny.

The findings are novel and potentially clinically important, and to my eyes this represents some of the most promising work in breath analysis yet undertaken.

The main limitation is that clearly these findings need to be prospectively validated in an independent study (ideally conducted by a different group) before the clinical implementation that is suggested in the conclusion.

It is also a shame that the protocol was not published - without that it is not possible to verify that the multiple subgroup comparisons [Aetiology x 3 groups, Outcomes 3 groups (clinical response, EEG response, s/e) x 2 categories (y/n)] were defined a priori, or instead represent selected findings from a retrospective exploratory data analysis. I wonder if the protocol could be added in the online supplement?

Stephen Fowler

Reviewer #3 (Remarks to the Author):

The paper by Singh et al. (COMMSMED-21-0055-T) describes the application of breath analysis to epileptic patients, assessing systemic drug concentrations along with risk estimates for drug response and side effects. The breath metabolome analysis was carried out by SESI-HRMS. The paper offers an interesting extension of breathomics to epilepsy; however it presents some pitfalls that need to be addressed.

1. Patients

The enrollment is quite "flexible". The patients are all unselected hospital outpatients. It is true that they represent "real-life" patients, but this impacts on the calculated models. Furthermore, the patients cover a "wide range of clinical presentations and pharmacotherapies". Are the authors sure that the covariates do not affect the results?

I would suggest that the authors present the data clearly stating that these are preliminary results and that this is a proof-of-concept study, starting from the title that should be changed accordingly.

2. Chemical identification

Since, "chemical identification of endogenous metabolites associated to non-responders and severe side effects could only be postulated", and "Chemical identification with highest degree of confidence would require further UPLC-MS/MS analysis using chemical standards, as done in this work for VPA metabolites" the discussion seems to be too detailed.

3. Text

Considering the above comments on the patients' enrollment and the chemical identification, I believe that the authors have put too much emphasis on their findings. It seems to me that the Results and Discussion sections describe too many details that could be avoided. That is, the biochemical descriptions should be reduced.

4. Limitations

The authors are aware of the limitations because the latter are clearly described on page 11. I would suggest to reduce the text (mainly the Discussion), putting forward the usefulness of the method, with its real-time application and its noninvasive approach. Overall, a better selection of patients (from the statistical and modeling point of views) is in progress in the lab (page 11, line

327) and will be reported in the near future.

In conclusion, the paper describes an interesting application of metabolomics to "real-life" epileptic outpatients, but it needs to be presented as a proof-of-concept study.

Reviewer #1 (Remarks to the Author):

1. *The present study is very relevant for improving clinical care of both children and adults with epilepsy suggesting a new method of non-invasive real time breath metabolomics to assess i) drug (specifically valproic acid) related metabolites and furthermore ii) drug regulated metabolites. I would categorize the present manuscript as a proof-of-concept study opening a possibility for a clinical breakthrough in evaluating individualized outcomes to a given anti-seizure medication (ASM – nowadays the preferred term). Naturally, much further research is needed to translate this new concept of measuring individualized effects with the help of this novel technique of not just valproic acid but other important ASM to everyday clinic, but the promise is definitely there and well enough backed by relevant analysis in this manuscript.*

These encouraging comments are much appreciated. I fully agree on the proof-of-concept categorization. I can anticipate that we are already validating the concept in cooperation with the largest epilepsy hospital in Switzerland. We hope to report on this large-scale study in the coming years.

We have also replaced the term “antiseizure drug (ASD)” to “antiseizure medication (ASM)” throughout the manuscript.

2. *However, I would like to suggest to improve the line of reasoning both in the introduction as well as in discussion in order to better convey the full potential of this novel approach. The deficiencies in present day conventional therapeutic drug monitoring (TDM) should be made more clear. The lack of clinical correlation between the efficacy outcomes or side-effects with ASM concentrations really decreases the value of TDM due to huge inter- and intra-individual variation, and many clinicians use TDM sparingly for this reason only in specific circumstances (pregnancy, known pharmacokinetic interactions etc.). In this case the golden standard is really not so golden, but of course the predictive performance of the novel approach should be compared with the present standard. I don't think the need to correlate ASM effects with metabolic pathways is less important in adults compared with children.*

We appreciate this remark, which we fully endorse. We have emphasized this problem in the following new paragraph using some excerpts of the referee's comment itself (hope you do not mind, please let us know otherwise and we will reformulate the paragraph).

“Perhaps even more importantly, conventional TDM presents fundamental shortcomings. The lack of clinical correlation between the efficacy outcomes or side-effects with ASM concentrations due to high inter- and intra-individual variation, decreases the value of TDM. This limitation discourages practitioners to use TDM, except perhaps in specific circumstances (e.g. pregnancy, known pharmacokinetic interactions etc.). The limitation that keeping the drugs' concentrations in the therapeutic range, does not assure an optimal clinical response of no side effects was also evidenced in our cohort. For example, in 32% of the paediatric measurements (40 out of 123), the patients were not responding to the therapy. However, of these 40 measurements, only in 18 at least one drug was outside its reference range.”

3. *The discussion about results concerning drug related metabolites and drug regulated metabolites should be dealt in a more precise way. Drug related metabolites are more straightforward and the emphasis on free concentrations of valproic acid should be made*

even more pronounced. The possibility of 3-heptanone and 4-OH-gamma-lactone with other predictors of blood valproic acid concentration to create new reference values independent of traditional blood reference values should be also explored. If any data from the present study or previously published studies is available, it would be interesting to have a short paragraph discussing the potential of this novel technique for measuring other ASM with this same approach.

We believe that this point was addressed in the following statement:

“For example, based on 3-heptanone and 4-OH- γ -lactone signal intensity ratios, we could estimate inter- and intra-individual variability in the activity of β and ω_1 -oxidation pathways of VPA metabolism. We found significant heterogeneity of metabolic pathway patterns within the sampled population and even within subjects during the course of this study (Supplementary Fig. 4B).”

We would prefer to be conservative and not to stretch more the potential of the drug-related metabolites might have to better phenotype patients receiving VPA. We are not aware of other examples of ASM, but we cite other related work, such as the recent detection of an antidepressant (venlafaxine) in breath of animal models. I can anticipate that we are currently working on AI regression models to predict ASM in patients and hope to report on this soon.

4. The data about correlations between side-effects and efficacy outcomes with specific metabolites is exciting and opens a really important scenario for future clinical praxis. The possibility of measuring specific metabolites with predicting power for side-effects and efficacy would be a significant improvement truly revolutionizing epilepsy treatment paradigm. However, there is also a possibility that other factors besides drug regulated pathways have an effect on the concentrations of these different metabolites. Correlation does not necessary mean causation, individual differences in for instance gut-bacteria regulated amino acids concentrations may cause some individuals to have either high or low systemic or CNS concentrations prior to starting valproic acid and therefore the effect may depend on an individual's prior state to a varying degree. Therefore, new studies where the amino acid or other relevant molecule concentrations are measured prior to starting a new ASM or other therapies such as neuromodulation (VNS or DBS). This approach to my understanding is not just related to ASM regulation but has also other applications and should be also discussed.

We fully agree. We tried to be cautious interpreting these results in the original version. We also admitted in the limitations section that:

” Third, the reported altered pathways in non-responders and patients suffering from side effects could not be completely tested in an independent cohort.”

We therefore did not make additional changes. We however anticipate that we continue investigating this concept of using these metabolites to provide risk scores. We are working at two levels: i) increasing the sample size and validating independently the predictions and ii) improving the classification mathematical models (i.e. AI).

5. The clinical data in present manuscript is well presented, and the inclusion of both adult and pediatric population is one of the strengths of the study. The limitations are well discussed.

We celebrate that you liked the overall presentation. We indeed put quite a lot of effort to present our study properly.

Reviewer #2 (Remarks to the Author):

1. *This very well written manuscript describes a detailed and original investigation into the real-time breath profile of children and adults with epilepsy, with the aims of discovering potential breath biomarkers of drug exposure and efficacy, and also disease control (on symptoms and EEG) and drug side effects. The findings of a potential breath signature of valproate (levels and efficacy) are very intriguing and certainly merit further study.*

Very much appreciated.

2. *I am not an expert in analytical chemistry or multivariate data analysis, but within these limitations I feel the methods and study design are sound and appropriate. I applaud the investigators on making the data open to independent scrutiny.*

We indeed made our best to be as rigorous as possible with our statistical and chemical analyses. We look forward for other investigators profiting from the data being publicly available. On a side note, as the referee is an expert in breath analysis, we feel that the credibility of the field would largely increase if we could further promote this practice within our community.

3. *The findings are novel and potentially clinically important, and to my eyes this represents some of the most promising work in breath analysis yet undertaken.*

Thank you for the praise. Coming from a colleague in the field, this is very encouraging.

4. *The main limitation is that clearly these findings need to be prospectively validated in an independent study (ideally conducted by a different group) before the clinical implementation that is suggested in the conclusion.*

We agree, however we feel that the prediction of USZ data (i.e. independent research group, machine, operator, etc.) included in this study is very reassuring. We expect SESI-HRMS technology to be deployed in more clinical settings in the future and hopefully this will be validated completely independently. I can anticipate that we are already validating the concept in cooperation with the largest epilepsy hospital in Switzerland. We hope to report on this large-scale study in the coming years.

5. *It is also a shame that the protocol was not published - without that it is not possible to verify that the multiple subgroup comparisons [Aetiology x 3 groups, Outcomes 3 groups (clinical response, EEG response, s/e) x 2 categories (y/n)] were defined a priori, or instead represent selected findings from a retrospective exploratory data analysis. I wonder if the protocol could be added in the online supplement?*

This is a very valid point. The further subgroup comparisons arose retrospectively when it became largely apparent the lack of clinical correlation between the efficacy outcomes or side-effects with ASM concentrations. We then wondered whether any of the other metabolites detected in exhaled breath could potentially serve as more accurate predictors than the drug concentration itself. For full transparency, now we enclose the clinical protocol as supplementary information.

Reviewer #3 (Remarks to the Author):

The paper by Singh et al. (COMMSMED-21-0055-T) describes the application of breath analysis to epileptic patients, assessing systemic drug concentrations along with risk estimates for drug response and side effects. The breath metabolome analysis was carried out by SESI-HRMS.

The paper offers an interesting extension of breathomics to epilepsy; however it presents some pitfalls that need to be addressed.

1. Patients

The enrollment is quite “flexible”. The patients are all unselected hospital outpatients. It is true that they represent “real-life” patients, but this impacts on the calculated models. Furthermore, the patients cover a “wide range of clinical presentations and pharmacotherapies”. Are the authors sure that the covariates do not affect the results?

I would suggest that the authors present the data clearly stating that these are preliminary results and that this is a proof-of-concept study, starting from the title that should be changed accordingly.

The reason to go for unselected patients was to test to what extent the concept of predicting blood concentrations of ASMs could be generalized, independently of the wide range of clinical presentations and pharmacotherapies. Perhaps the model would have been more accurate predicting blood concentrations based on the breath metabolites if we would have selected for example monotherapy patients only. On the other hand, this would lessen the value of the proposed technique because of the limitations to be applicable to a rather reduced set of patients. In this regard, we believe that the results shown in Figure 4 clearly show that, although there is room for improvement (i.e. decreased RMSE), the model predicts reasonable well systemic blood concentrations in two independent heterogenous datasets.

We are aware that this heterogeneity has an impact in the way the patients metabolize the drugs. Indeed, this the main motivation of this study, as there is an unmet clinical need to better stratify the very high intra- and inter-patient variability in responding to medication. For example, as show in Supplementary Fig. 2 B-D, and in full agreement with the literature, we observe a quadratic dependency between free vs total VPA, a negative correlation of free VPA with age and a higher total VPA serum concentration at a given dose for monotherapy patients than for multitherapy. The question here is: are these covariates influencing dramatically our research question? We believe that, based on the evidence presented, the answer is no. Obviously, this heterogeneity has an impact in the circulating free VPA (Supplementary Fig. 2 B-C), but what we tried to address here was: given a certain blood concentration of VPA, is there a relationship with the concentration of its exhaled metabolites that allow for a prediction of the blood concentration? Again, we believe that Fig. 4 clearly shows that the answer is yes, although it is true that this heterogeneity may contribute to the prediction errors that we observe.

Regarding the request of emphasizing (even more) the proof-of-concept nature of the study, we fully agree. We have clearly stated already at the abstract level that this is a proof-of-principle study. We also include as supplementary information the clinical protocol of the study, which is entitled: “Feasibility of Therapeutic Drug Monitoring Guided by Exhaled Breath Analysis: A pilot study”. If the editor requests it, we are happy to emphasize the feasibility nature of the study even at the title level.

2. Chemical identification

Since, “chemical identification of endogenous metabolites associated to non-responders and severe side effects could only be postulated”, and “Chemical identification with highest degree of confidence would require further UPLC-MS/MS analysis using chemical standards, as done in this work for VPA metabolites” the discussion seems to be too detailed.

We believe that we have openly acknowledged the different degrees of certainty in the chemical assignments. While we clearly admit that the identification of the endogenous metabolites is based on accurate masses and subsequent query to metabolomics databases, it is also true that we used a stringent and widely accepted computational approach (i.e. enrichment analysis, which is used across all -omics platforms) to identify alterations at the metabolic pathway level, rather than at the individual molecule level. In particular, we used GSEA and Mummichog algorithms. The excerpt below from Shuzhao Li (ed.), Computational Methods and Data Analysis for Metabolomics, Methods in Molecular Biology, vol. 2104, https://doi.org/10.1007/978-1-0716-0239-3_17 emphasizes our argument:

“Due to this challenge, functional interpretation of global metabolomics data is not straightforward, as classical metabolic pathway enrichment analysis requires metabolites as input, not MS peaks. To address this bottleneck, Li et al. proposed a novel approach, named mummichog, to directly infer pathway activities from peak lists by leveraging the collective power of metabolic pathways, without requiring a priori metabolite identification [20]. **This algorithm assumes that a certain degree of random errors during individual peak assignment will not change the collective behavior jointly determined by all metabolites involved in the pathways.** This concept has been recently adapted to the popular Gene Set Enrichment Analysis (GSEA) algorithm [21] in MetaboAnalystR 2.0 and MetaboAnalyst 4.0. Mummichog is based on over representation analysis (ORA) to test if certain pathways are enriched in the significant peaks as compared to null models based on peak lists of the same size randomly drawn from the inputted peak list.”

Along the same line in Metabolites **2019**, 9, 57; doi:10.3390/metabo9030057”

“[...] a key concept is to shift the unit of analysis from individual compounds to individual pathways or a group of functionally related compounds (i.e., metabolite sets [17]). The general assumption is that the collective behavior of a group is more robust against a certain degree of random errors of individuals”

In conclusion, we have focused the discussion at this high level (i.e., significantly altered pathways; Fig. 5), and we refrained of making claims from any specific metabolite or pathways not significantly altered after the enrichment analysis.

3. Text

Considering the above comments on the patients’ enrollment and the chemical identification, I believe that the authors have put too much emphasis on their findings. It seems to me that the Results and Discussion sections describe too many details that could be avoided. That is, the biochemical descriptions should be reduced.

As discussed above, we have phrased our biological interpretation rather conservatively at the pathway level based on the enrichment analysis, which accounts for random errors during individual peak assignment. It would be very difficult to reduce the discussion without conveying the message stemming from the enrichment analysis.

4. *Limitations*

The authors are aware of the limitations because the latter are clearly described on page 11. I would suggest to reduce the text (mainly the Discussion), putting forward the usefulness of the method, with its real-time application and its noninvasive approach. Overall, a better selection of patients (from the statistical and modeling point of views) is in progress in the lab (page 11, line 327) and will be reported in the near future.

In conclusion, the paper describes an interesting application of metabolomics to “real-life” epileptic outpatients, but it needs to be presented as a proof-of-concept study.

In conclusion, we have stressed the proof-of-concept nature of the study in this revised version.

Reviewers' comments:

Reviewer #1 (Remarks to the Author):

I am satisfied with the answers provided by the authors.

Reviewer #2 (Remarks to the Author):

Thank you for addressing my comments, and in particular for adding the protocol. I have no further comments

Reviewer #3 (Remarks to the Author):

In essence, the revised version of the manuscript answers the questions raised, at least from the philosophical point of view.

I was expecting more "numerical" details, in particular for point 1. I can understand that "what we tried to address here was: given a certain blood concentration of VPA, is there a relationship with the concentration of its exhaled metabolites that allow for a prediction of the blood concentration? Again, we believe that Fig. 4 clearly shows that the answer is yes, although it is true that this heterogeneity may contribute to the prediction errors that we observe." but, still, that covariates do not affect the model should have been evaluated.

Similarly, for the chemical identification point. I understand that "This algorithm assumes that a certain degree of random errors during individual peak assignment will not change the collective behavior jointly determined by all metabolites involved in the pathways.", however, the question remains: Why the discussion is so detailed if the details are (statistically) guessed?

Finally, why "It would be very difficult to reduce the discussion without conveying the message stemming from the enrichment analysis."? The idea is that the message should present a word of caution because metabolomics should avoid speculation on the data.

Reviewer #3 (Remarks to the Author):

In essence, the revised version of the manuscript answers the questions raised, at least from the philosophical point of view.

I was expecting more "numerical" details, in particular for point 1. I can understand that "what we tried to address here was: given a certain blood concentration of VPA, is there a relationship with the concentration of its exhaled metabolites that allow for a prediction of the blood concentration? Again, we believe that Fig. 4 clearly shows that the answer is yes, although it is true that this heterogeneity may contribute to the prediction errors that we observe." but, still, that covariates do not affect the model should have been evaluated.

I understand the reviewer's point. We have now re-run the regression model using the covariates (i.e. gender, age and number of frugs) as additional predictors. The results are shown in the new supplementary figure 11. The associated text in the main manuscript reads as follows:

“Interestingly, we observed that including covariate such as age, gender and number of ASMs as predictors does not necessarily make the VPA prediction any better (Supplementary Fig. 11E-F). Hence, to reduce the complexity, we used the model with only exhaled VPA-related ions as predictors.”

Please note that in this revised version we have used Lin's concordance correlation coefficient as a metric for model assessment because it is dimensionless, hence makes it more convenient to compare predictions across different scales (i.e. free and total VPA).

Similarly, for the chemical identification point. I understand that "This algorithm assumes that a certain degree of random errors during individual peak assignment will not change the collective behavior jointly determined by all metabolites involved in the pathways.", however, the question remains: Why the discussion is so detailed if the details are (statistically) guessed?

Finally, why "It would be very difficult to reduce the discussion without conveying the message stemming from the enrichment analysis."? The idea is that the message should present a word of caution because metabolomics should avoid speculation on the data.

I agree that over-interpretation should be avoided. For this reason, we have extended the limitations statement as follows:

“Second, chemical identification of endogenous metabolites associated to non-responders and side effects could only be postulated based on the database matching of measured accurate masses (within 2 ppm). Chemical identification with highest degree of confidence would require further UPLC-MS/MS analysis using chemical standards, as done in this work for VPA metabolites. For this reason, although the enrichment analysis algorithm used here analyses at a collective level the behaviour of groups of metabolites (assuming random errors at the individual peak level), the biochemical/metabolic interpretation should be taken cautiously until unambiguous chemical identification is provided.”

Please note that, meanwhile, we have been working on the identification of the most relevant metabolites. I enclose here preliminary results confirming that some of the most important

assignments (i.e. Tyrosine, GABA and glutamic acid) are correct (ongoing project; data for review only). I hope this instils further confidence on the arguments made in this manuscript.

[IMAGE REDACTED]

REVIEWERS' COMMENTS:

Reviewer #3 (Remarks to the Author):

All comments have been addressed.